# Oops, I Sampled it Again: Reinterpreting Confidence Intervals in Few-Shot Learning

**Raphael Lafargue**                                        *raphael.lafargue@imt-atlantique.fr*
*IMT Atlantique, Lab-STICC, UMR CNRS 6285, F-29238, France*
*Australian Institute for Machine Learning, University of Adelaide, Australia*
*CNRS, IRL CROSSING, Adelaide, Australia*

**Luke Smith**                                              *luke.a.smith@adelaide.edu.au*
*Australian Institute for Machine Learning, University of Adelaide, Australia*

**Franck Vermet**                                           *franck.vermet@univ-brest.fr*
*LBMA, CNRS, UMR 6205, Univ Brest, Brest, France*

**Matthias Löwe**                                           *matthloewe@googlemail.com*
*University of Münster, Germany*

**Ian Reid**                                                *ian.reid@adelaide.edu.au*
*Australian Institute for Machine Learning, University of Adelaide, Australia*
*MBZUAI, Abu Dhabi, United Arab Emirates*
*CNRS, IRL CROSSING, Adelaide, Australia*

**Vincent Gripon**                                          *vincent.gripon@imt-atlantique.fr*
*IMT Atlantique, Lab-STICC, UMR CNRS 6285, F-29238, France*

**Jack Valmadre**                                           *jack.valmadre@adelaide.edu.au*
*Australian Institute for Machine Learning, University of Adelaide, Australia*
*CNRS, IRL CROSSING, Adelaide, Australia*

**Reviewed on OpenReview:** *https://openreview.net/forum?id=JxxkKt9yrx*

## Abstract

The predominant method for computing confidence intervals (CI) in few-shot learning (FSL) is based on sampling the tasks with replacement, i.e. allowing the same samples to appear in multiple tasks. This makes the CI misleading in that it takes into account the randomness of the sampler but not the data itself. To quantify the extent of this problem, we conduct a comparative analysis between CIs computed with and without replacement. These reveal a notable underestimation by the predominant method. This observation calls for a reevaluation of how we interpret confidence intervals and the resulting conclusions in FSL comparative studies. Our research demonstrates that the use of paired tests can partially address this issue. Additionally, we explore methods to further reduce the (size of the) CI by strategically sampling tasks of a specific size. We also introduce a new optimized benchmark, which can be accessed at `https://github.com/RafLaf/FSL-benchmark-again`.

## 1 Introduction

The recent surge of interest in few-shot learning (FSL), driven by its potential applications in many real-world scenarios, has led to a proliferation of new methods and novel experimental protocols (Sung et al., 2018; Snell et al., 2017; Bendou et al., 2022; Zhang et al., 2020). If in conventional machine learning it is common to benchmark methods using a fixed split into training and validation sets, FSL presents unique

challenges due to its reliance on extremely small and, consequently, biased training datasets. In fact, the performance of FSL can dramatically depend on the choice of the given labeled training samples (Arnold et al., 2021).

One question that FSL shares with conventional machine learning is that of the best performing methods. Especially relevant to FSL, the high variance of measured performance based on the choice of labeled data has led practitioners to quickly adopt the standard of aggregating statistics over a large number of artificially generated tasks, stemming from a single (or a few distinct) dataset(s). The predominant approach is to generate artificial few-shot tasks by randomly sampling the same dataset with replacement, i.e. permitting the same samples to appear across multiple tasks. The outcome of these numerous tasks is the calculation of an average accuracy and its associated confidence interval (CI) for each method, thereby providing researchers with a statistically relevant basis for comparing the efficacy of different methods.

By allowing the same samples to appear in multiple tasks, the computed CIs account for the randomness of the sampler but not the data itself. In fact, the computed CIs use the usual Lindeberg-Lévy Central Limit Theorem (CLT). Hence, for these CIs to be statistically valid, the underlying random variables must be independent and identically distributed (IID). This means that the currently reported CIs should be understood as a likely range of outcomes if the experiment were reproduced using *exactly the same data*, which we will refer to in the remainder of this paper as **Closed CIs (CCIs)**. This contrasts with what is often of interest in many areas of machine learning: the range of outcomes if the experiment were repeated with data from the same *underlying distribution*, termed **Open CIs (OCIs)**. Note that the latter could be obtained simply by sampling tasks without replacement, but at the cost of considerably restricting the number of different tasks one can generate for a given dataset. This limitation is even more severe if the dataset is small, resulting in potentially larger CIs and inconclusive comparisons between methods.

The purpose of this paper is to highlight this crucial consideration when computing CIs. We propose strategies to address this issue and obtain meaningful comparisons while still accounting for the randomness of the data. These strategies rely on a) **Paired Tests (PT)**, where methods are evaluated on the same set of generated tasks, and b) adequately sizing tasks. Throughout the paper, we focus on the specific case of few-shot classification in vision, the most popular area of research in the field of FSL.

Our investigation using Open Confidence Intervals (OCIs) can lead to conclusions that are inconsistent with those obtained using the classical approach in the field of few-shot learning. In particular, we find that some methods previously reported as statistically significantly outperforming others are actually indistinguishable when using OCIs, and vice versa. In addition, we show cases where the use of CCIs lead to (statistically significant) conclusions that are diametrically opposed to those obtained using PT. One such example is given

|  |  | CLIP | | DINO | |
| --- | --- | --- | --- | --- | --- |
|  |  | NCC | FT | NCC | FT |
| CLIP | NCC |  | 0 0 0 | 0 + + | − **0** + |
|  | FT | − 0 0 |  | 0 + + | 0 0 + |
| DINO | NCC | + 0 0 | + 0 + |  | 0 0 − |
|  | FT | + 0 0 | + 0 0 | 0 0 0 |  |

Table 1: Comparison between different methods for few-shot classification. Each entry consists of three elements: [**With Replacement (Closed), Without Replacement (Open), Paired Tests (PT)**]. Symbols + and − respectively indicate significant differences (positive and negative) between the row method and the column method while 0 is non-conclusive. Results derive from the DTD test split (bottom left triangle) and Traffic Signs (top-right) split of MetaDataset, with task sampling at 5 shots, 5 ways, and 15 queries. **Note the inversion in bold**. NCC (Nearest Class Centroid), FT (Fine-tune) with CLIP and DINO as feature extractors.

in Table 1, where we compare different methods for few-shot classification depending on their used feature extractor (here CLIP (Radford et al., 2021) or DINO (Caron et al., 2021)) and their adapting methods (here Logistic Regression (LR), Nearest Class Centroid (NCC) or Fine-Tuning (FT)). In the table, we report three conclusions for each pair of model and method combinations: the first is obtained from the methodology described in Luo et al. (2023) where the authors used the predominant way to compute CIs, that is CCIs; the second OCIs, and the third is based on PT. A conclusion that the row method is better than the column

one is denoted with +, 0 means there is no conclusive statement, and – that the column method is the most performing of the two. The upper triangular values in blue correspond to the Traffic Sign test split while the rest of the table (in red) refers to the DTD test split in the Metadataset benchmark by Triantafillou et al. (2019). Particularly striking results are found on the Traffic Signs dataset. Indeed, when studied with replacement tests, CLIP with the NCC adapter underperforms DINO with Fine-tuning, yet this outcome is reversed in paired test assessments. This clearly demonstrates the importance of distinguishing between measurement methods, as failing to do so can lead to significant misinterpretations of results.

The main contributions of this work are:

- We highlight the importance of considering data replacement when computing CIs for FSL methods comparison. Our study illustrates the impact on CI ranges when transitioning from closed to open CIs on standard off-the-shelf vision datasets.

- By implementing paired evaluations, wherein multiple methods are compared on identically generated task sets, we demonstrate the ability to reach conclusive comparisons more frequently than when relying on simple performances with OCIs.

- We investigate how to optimize task generation from a given dataset, taking into account its size and number of classes, to reach small ranges of CIs and lead to more conclusive comparisons between methods. The result is a benchmark that can be used for few-shot classification of images.

## 2 Closed CIs vs. Open CIs

In this section, we are interested in better quantifying the difference between CCIs and OCIs. For this purpose, we define notations and outline algorithms for task sampling. We present a theoretical analysis of the differences between CCIs and OCIs. Finally, we empirically compare their ranges on real datasets.

### 2.1 A mathematical description of the problem

#### 2.1.1 Standard Evaluation and Notations

The predominant method of evaluation in the field of few-shot classification is described in Algorithm 1. A few-shot classification task $\mathcal{T} = (\mathcal{K}, \mathcal{S}, \mathcal{Q})$ comprises a set of classes $\mathcal{K}$, a support set $\mathcal{S} = \{\mathcal{S}_c\}_{c \in \mathcal{K}}$ and a query set $\mathcal{Q} = \{\mathcal{Q}_c\}_{c \in \mathcal{K}}$ where $\mathcal{S}_c, \mathcal{Q}_c$ denote the sets of support and query examples for each class $c \in \mathcal{K}$. Let $K = |\mathcal{K}|$ denote the number of *ways* (i.e. classes in a few-shot task), $S = |\mathcal{S}_c|$ the number of *shots* per class, and $Q = |\mathcal{Q}_c|$ the number of *queries* per class (for simplicity, we assume the classes to be balanced). Few-shot evaluation is typically performed by constructing many tasks from a larger evaluation dataset. An evaluation dataset $\mathcal{D} = (\mathcal{C}, \mathcal{X})$ comprises a set of classes $\mathcal{C}$ and examples for all classes $\mathcal{X} = \{\mathcal{X}_c\}_{c \in \mathcal{C}}$. Let $C = |\mathcal{C}| \geq K$ denote the number of classes, and $N = |\mathcal{X}_c| \gg S + Q$ the number of examples per class. As highlighted in the introduction, the rationale followed by the CI computation predominant method is to consider $\mathcal{D}$ to be fixed and non-probabilistic.

---

**Algorithm 1** Predominant evaluation algorithm

---

1: **procedure** EVALUATE($T, K, S, Q, \mathcal{C}, \{\mathcal{X}_c\}_{c \in \mathcal{C}}$)  ▷ $T$ tasks, $K$ ways, $S$ shots, $Q$ queries, set of classes $C$, set of data samples $\{\mathcal{X}_c\}_{c \in \mathcal{C}}$
2:    **for** $t = 1, \ldots, T$ **do**
3:        $\mathcal{K} \leftarrow \text{take}(K, \text{shuffle}(\mathcal{C}))$
4:        **for** $c \in \mathcal{K}$ **do**
5:            $\mathcal{S}_c, \mathcal{Q}_c \leftarrow \text{split}(S, \text{take}(S + Q, \text{shuffle}(\mathcal{X}_c)))$
6:        **end for**
7:        $f_{\mathcal{S}} \leftarrow \text{FEWSHOTLEARN}(\mathcal{S})$                                      ▷ $\mathcal{S} := \{\mathcal{S}_c\}_{c \in \mathcal{K}}$
8:        $\bar{A}_t \leftarrow \frac{1}{KQ} \sum_{c \in \mathcal{K}} \sum_{x \in \mathcal{Q}_c} \mathbb{1}[f_{\mathcal{S}}(x) = c]$
9:    **end for**
10:    $\bar{A} \leftarrow \text{MEAN}(A)$                                      ▷ $\text{MEAN}(A) := \frac{1}{T} \sum_{t=1}^{T} A_t$
11:    $\sigma_A \leftarrow \sqrt{\text{VAR}(A)}$                              ▷ $\text{VAR}(A) := \frac{1}{T-1} \sum_{t=1}^{T} (A_t - \bar{A})^2$
12:    $\sigma_{\bar{A}} \leftarrow \sigma_A / \sqrt{T}$
13:    **return** $\bar{A} \pm 1.96 \sigma_{\bar{A}}$                          ▷ $1.96 \sigma_{\bar{A}} = r(p_{limit} = 95\%) \sigma_{\bar{A}}$
14: **end procedure**

---

The standard few-shot task sampler constructs $T$ random tasks with $K$ ways, $S$ shots and $Q$ queries from a dataset $\mathcal{D}$ as outlined in Algorithm 1. This procedure introduces additional random variables (besides the dataset itself) in the selection of classes and examples. Let $\mathcal{T}_t = (\mathcal{K}_t, \mathcal{S}_t, \mathcal{Q}_t)$ for $t = 1, \ldots, T$ denote the sampling of tasks. The average accuracy on each task is obtained as:

$$A_t = \frac{1}{KQ} \sum_{c \in \mathcal{K}_t} \sum_{x \in \mathcal{Q}_{tc}} \mathbb{1}[f_{\mathcal{S}_t}(x) = c], \tag{1}$$

where $f$ is the evaluated model, conditioned by the support set.

Next, we turn to the actually reported metric which is the average accuracy over several tasks.

### 2.1.2 Computing Confidence Intervals

We compute the accuracy $A_t$ of the method on each task, and then take the mean across tasks $\bar{A} = \frac{1}{T}(A_1 + \cdots + A_T)$. As such, we obtain the following formula for the variance:

$$\text{Var}[\bar{A}] = \frac{\text{Var}[A_1] + \cdots + \text{Var}[A_T]}{T^2} = \frac{\text{Var}[A]}{T} . \tag{2}$$

Assuming a sufficient sample size and that the mean $\bar{A}$ is normally distributed according to the Central Limit Theorem, the 95% confidence interval is obtained as $\bar{A} \pm 1.96 \sigma_{\bar{A}}$ using the formula for standard error (standard deviation of the sample mean):

$$CI = 1.96 \sigma_{\bar{A}} = 1.96 \frac{\sigma_A}{\sqrt{T}} . \tag{3}$$

Note that in the case of a very small number of tasks, Student's distributions can be used instead. Also, we took the example of 95% CIs, which is arbitrary but very common in the literature. For more generality, we consider a probability $p_{limit}$ in the following for all theoretical considerations, and stick with the 95% value for experiments.

Note that, as the number of tasks becomes larger, it is inevitable that many tasks will re-use examples, since tasks are constructed independently *with* replacement. Therefore, as $T$ becomes large, the variance of the sample mean will approach the *conditional* variance $\text{Var}[\bar{A} \mid \mathcal{D}]$, and the confidence interval will represent the likely range of outcomes if we were to repeat the experiment with a random set of tasks on the *same*

*dataset.* In other words, it provides no insight into how well a method would generalize on a distribution. On the other hand, if $T$ is chosen small enough, there will be minimal re-use of examples and the assumption of independence may approximately hold, although the confidence interval may be significantly larger.

We will now conduct an empirical evaluation of the disparity between closed (tasks sampled with replacement) and open (tasks sampled without replacement) CIs on real datasets.

## 2.2 Are OCIs larger than CCIs? An empirical study

In contrast to Algorithm 1, the task sampling without replacement is presented in Algorithm 2 (see Appendix). Note that we make an explicit use of a Student's law estimator in this algorithm as it is expected that for some small datasets it can only generate but a few independent tasks. In the latter, the total number of tasks $T$ is determined directly by the sampling process, thanks to a specific stopping condition, based on the exhaustion of the dataset. This is done in an effort to minimize the obtained CIs' ranges. Since samples cannot be sampled twice, we can consider that the classes and examples are drawn IID from an underlying data distributions $p(C)$ and $p(X \mid C)$. This is why OCIs also account for the randomness of the data.

In our experiments, we utilize datasets from the Metadataset Benchmark as referenced in Triantafillou et al. (2019). This benchmark comprises 10 datasets, out of which we employ 9, excluding Imagenet, to focus on cross-domain results in line with the recent trend in the literature (Zhou et al., 2022b). These include Omniglot (handwritten characters), Aircraft, CUB (birds), DTD (textures), Fungi, VGG Flowers, Traffic Signs, Quickdraw (crowd-sourced drawings) and MSCOCO (common objects) (Lake et al., 2015; Maji et al., 2013; Wah et al., 2011; Cimpoi et al., 2014; Schroeder & Cui, 2018; Nilsback & Zisserman, 2008; Houben et al., 2013; Jongejan et al., 2016; Lin et al., 2014).

Luo et al. (2023) details few-shot accuracies for 2000 tasks with 5-shots, 5 ways, and 15 queries in a comprehensive table covering various works on the Metadataset datasets. Our study's only difference lies in the adoption of the $T = 600$ setting, a more prevalent choice in existing literature. If CCIs are found to be narrower than OCIs with this smaller $T$, it will be even starker with $T = 2000$ tasks as shown in Equation 3. Our primary reference for methods and models is the comprehensive compilation provided by Luo et al. (2023), a foundational starting point for our experiments.

Our findings are detailed in Table 2, showcasing results across different few-shot methods and datasets. Firstly, there is a noticeable homogeneity in CCIs, arising from the fixed number of tasks set at $T = 600$, which contrasts with the variability observed in OCIs. Interestingly, CCIs are substantialy narrower than OCIs for small datasets such as Aircraft and DTD. Conversely, in the case of larger datasets like Quickdraw, CCIs become larger than OCIs due to $T = 600$ being insufficient to deplete the dataset. Indeed, Aircraft and DTD's test splits contain 1,500 and 840 samples respectively, whereas the test splits for MSCOCO and Quickdraw have much larger sizes of 152,000 and 7.7 million samples respectively. Across various datasets, models and methods, CCIs are on average 3.8 times larger than OCIs. These results highlight the imperative need for accurate interpretation of Confidence Intervals, given the dramatic differences between OCIs and CCIs ranges, that undoubtedly lead to disagreeing conclusions if misinterpreted.

We also notice that for cases where methods reach accuracies near 100%, like adaptation methods using CLIP (unlike those using DINO) on the CUB dataset, both types of CIs become narrower. This is due to accuracy saturation at 100%, which reduces the standard deviation of accuracies.

In the following, we delve into the conclusiveness of comparative studies using CCIs or OCIs.

## 2.3 Impact on Conclusiveness

First let us recall how confidence intervals are used to draw conclusions when comparing methods. Suppose we have two variables of interest $x_1$ and $x_2$, with their corresponding $p_{limit}$ - confidence intervals (a generalized version of 95%-CIs) $[\bar{x}_1 - \delta_1, \bar{x}_1 + \delta_1]$ and $[\bar{x}_2 - \delta_2, \bar{x}_2 + \delta_2]$. To draw conclusions about the fact $x_1$ is smaller than $x_2$, we proceed as follows: if the two intervals do not intersect, and $x_1 + \delta_1 < \bar{x}_2 - \delta_2$, then:

| Dataset Name | Dataset Size | Model Sampling Method | CLIP | | DINO | |
| --- | --- | --- | --- | --- | --- | --- |
| | | | W. Repl. | W/O. Repl. | W. Repl. | W/O. Repl. |
| DTD | 840 | LR | $79.59 \pm 0.52$ | $84.00 \pm 4.50$ | $83.29 \pm 0.51$ | $86.10 \pm 4.66$ |
| | | FT | $76.87 \pm 0.56$ | $80.76 \pm 5.60$ | $81.82 \pm 0.51$ | $84.19 \pm 5.76$ |
| VGG Flower | 1,425 | LR | $98.30 \pm 0.23$ | $99.39 \pm 0.84$ | $97.48 \pm 0.26$ | $97.58 \pm 2.07$ |
| | | FT | $98.33 \pm 0.23$ | $99.27 \pm 0.93$ | $97.13 \pm 0.29$ | $97.45 \pm 1.98$ |
| Aircraft | 1,500 | LR | $75.79 \pm 0.85$ | $69.90 \pm 5.59$ | $59.04 \pm 0.93$ | $53.62 \pm 5.66$ |
| | | FT | $74.70 \pm 0.85$ | $68.95 \pm 5.82$ | $54.58 \pm 0.94$ | $49.81 \pm 5.17$ |
| CUB | 1,770 | LR | $96.85 \pm 0.28$ | $97.24 \pm 1.88$ | $92.06 \pm 0.43$ | $89.69 \pm 4.03$ |
| | | FT | $96.68 \pm 0.29$ | $97.07 \pm 1.83$ | $89.16 \pm 0.52$ | $87.47 \pm 4.64$ |
| Omniglot | 13,180 | LR | $90.55 \pm 0.49$ | $91.04 \pm 0.89$ | $93.67 \pm 0.38$ | $94.12 \pm 0.79$ |
| | | FT | $92.06 \pm 0.48$ | $92.83 \pm 0.91$ | $94.70 \pm 0.36$ | $95.09 \pm 0.70$ |
| Fungi | 13,463 | LR | $70.86 \pm 0.88$ | $74.78 \pm 1.89$ | $77.26 \pm 0.75$ | $81.64 \pm 1.41$ |
| | | FT | $68.03 \pm 0.95$ | $71.87 \pm 1.96$ | $74.02 \pm 0.83$ | $78.88 \pm 1.68$ |
| Traffic Signs | 39,252 | LR | $82.99 \pm 0.87$ | $77.02 \pm 0.92$ | $83.63 \pm 0.92$ | $75.58 \pm 1.17$ |
| | | FT | $83.64 \pm 0.85$ | $76.69 \pm 0.96$ | $84.20 \pm 0.92$ | $74.93 \pm 1.23$ |
| MSCOCO | 151,545 | LR | $72.27 \pm 0.78$ | $67.97 \pm 0.63$ | $76.05 \pm 0.72$ | $72.02 \pm 0.60$ |
| | | FT | $70.22 \pm 0.79$ | $65.97 \pm 0.64$ | $75.18 \pm 0.75$ | $71.19 \pm 0.61$ |
| Quickdraw | 7,710,295 | LR | $75.79 \pm 0.66$ | $75.54 \pm 0.36$ | $74.54 \pm 0.68$ | $74.07 \pm 0.38$ |
| | | FT | $76.18 \pm 0.69$ | $75.93 \pm 0.38$ | $74.10 \pm 0.70$ | $73.61 \pm 0.39$ |

Table 2: Accuracies and associated CIs of methods on different datasets with and without replacement in the sampling of tasks. The dataset size correspond to the number of images in each test split. FT and LR respectively stand for Fine-tune and Logistic Regression.

$$P\left(x_1 < x_2\right) > P\left(x_1 < \bar{x}_1 + \delta_1 \wedge x_2 > \bar{x}_2 - \delta_2\right) = \left(1 - \frac{1 - p_{limit}}{2}\right)^2 > p_{limit} \, , \tag{4}$$

where the $(1 - p_{limit})/2$ part comes from the symmetry of the Gaussian distribution.

With this in mind, the results listed in Table 2 can lead to different conclusions depending on whether CCIs or OCIs are used. For example, using the DTD dataset, CCIs lead to conclusive comparisons between both backbones and methods, whereas OCIs are inconclusive as they all intersect. Again, this should not be seen as a contradiction, but rather as having different paradigms about the comparisons. CCIs compare methods if they were reused on the same data, whereas OCIs focus on the underlying distribution. Next, we propose two methods to improve conclusiveness when comparing methods, namely paired tests in Section 3 and task sizing in Section 4.

## 3 Paired tests

### 3.1 Definitions

As an effort to reduce the ranges of Open Confidence Intervals (OCIs), we propose to make use of paired tests. Indeed, as we pointed out in the introduction, FSL tasks have a vast diversity in difficulty, leading to a high variance in accuracy across tasks. It is noteworthy that a task deemed *hard* for method A often aligns in difficulty for method B. This parallel in task difficulty across different methods was

previously identified by Arnold et al. (2021), and our findings are in agreement, as illustrated in Figure 1, where we plot the accuracies on tasks generated from the Traffic Sign dataset and using two different combinations of feature extractors and adaptation method. In the provided figure, a strong correlation, quantified at 0.675, is evident between two distinct methods. This highlights the potential for reducing the variance in accuracies resulting from task sampling by employing paired testing.

Formally, let us denote $\Delta_t = A_t - B_t$ the accuracy of the method on task t relative to a method with accuracy $B_t$, with $\bar{\Delta} = \frac{1}{T} \sum_{t=1}^{T} \Delta_t$ the mean difference across tasks. While the mean of differences is simply the difference of the means, $\mathbb{E}[\bar{\Delta}] = \mathbb{E}[\bar{A}] - \mathbb{E}[\bar{B}]$, the variance of the difference may be significantly reduced when the accuracies are positively correlated:

$$\text{Var}[\bar{\Delta}] = \text{Var}\left[\frac{1}{T}\sum_{t=1}^{T}\Delta_t\right] = \frac{1}{T}\text{Var}\left[A_t - B_t\right] \quad (5)$$

since $\text{Var}[X - Y] = \text{Var}[X] + \text{Var}[Y] - 2\,\text{Cov}(X, Y)$.

The lower variance of $\Delta_t$ compared to $A_t$ results in a correspondingly smaller confidence interval, as detailed in Equation 3. Consequently, this leads to scenarios where two methods can exhibit significant differences when analyzed using paired testing despite there being no significant differences when directly comparing accuracies. We performed experiments comparing various methods to fine-tuning (FT). Results are shown in Table 3, where each line corresponds to a specific dataset and feature extractor

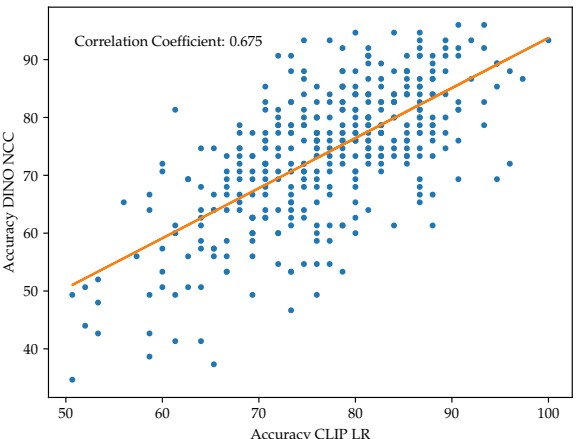

Figure 1: Scatter plot of task accuracies using two different combinations of feature extractor/adaptation methods on the Traffic Signs benchmark.

and each column to a combination of an adaptation method and a feature extractor. Two conclusions are depicted, the first being based on directly comparing accuracies, and the second using paired tests instead. Note that in all cases, we rely on OCIs, that is to say sampling without replacements. Let us also notice that paired tests never lead to contradictory conclusions with direct comparisons of accuracies, a direct consequence of previous remarks about the mean of differences. Their difference lies in the ability to conclude or not.

The fine-tuning adaptation method is selected as the baseline in Table 3, primarily due to its substantial computational cost. The cost of fine-tuning stems from the necessity to update the weights of the entire feature extractor for each considered task. The objective is to determine if it surpasses more cost-effective methods in performance. In comparative analyses excluding the comparison between CLIP and DINO and focusing only on the same feature extractor, it is frequently noted that fine-tuning either underperforms relative to other methods or the results are inconclusive. Indeed, fine-tuning significantly outperforms other methods in only 4 out of 36 cases, while underperforming in 14 cases. The outcomes in the remaining instances remain inconclusive. Consequently, we conclude that fine-tuning, in light of its considerable computational overhead, is not particularly advantageous to consider. These results should be nuanced by the dependency of fine-tuning on hyperparameters which makes any assertions about this method contingent upon a specific set of hyperparameters (Kumar et al., 2022).

For each of the nine considered datasets, we have a total of 135 unique comparisons, taking into account only distinct pairs of (model, methods) across two models and three methods. We found that 57 comparisons were conclusive using direct comparison with OCI while 94 were conclusive using *paired tests*. Figure 6 in the Appendix illustrates this.

Table 1 illustrates, on the DTD and Traffic signs dataset, that the three different approaches for computing CIs discussed so far result in varied assessments of the significance of differences between two methods. On all datasets, in 27 instances out of the 114 where the comparison with replacement was conclusive, that is $\sim 23\%$ of such cases, a pattern emerges: a comparison initially classified as significantly different becomes non-conclusive under sampling without replacement, and then conclusive again when a paired test is applied. On 12 instances, $\sim 11\%$ of previously considered cases, conclusiveness is not confirmed by the paired test.

| Model | Model Method Dataset | CLIP | | | DINO | | |
|---|---|---|---|---|---|---|---|
| | | LR | NCC | FT | LR | NCC | FT |
| CLIP | Aircraft | 00 | 00 | | ++ | ++ | ++ |
| | CUB | 00 | 00 | | ++ | ++ | ++ |
| | DTD | 0– | 00 | | 0– | 0– | 00 |
| | Fungi | 0– | 0– | | –– | –– | –– |
| | MSCOCO | –– | 0– | | –– | –– | –– |
| | Omniglot | 0+ | 00 | | 0– | –– | –– |
| | Quickdraw | 0+ | 00 | | ++ | ++ | ++ |
| | Traffic Signs | 00 | 00 | | 0+ | ++ | 0+ |
| | VGG Flower | 00 | 00 | | 0+ | 0+ | 0+ |
| DINO | Aircraft | –– | –– | –– | 0– | 00 | |
| | CUB | –– | –– | –– | 0– | 00 | |
| | DTD | 00 | 00 | 00 | 00 | 00 | |
| | Fungi | ++ | ++ | ++ | 0– | 0– | |
| | MSCOCO | ++ | ++ | ++ | 0– | 0– | |
| | Omniglot | ++ | ++ | ++ | 0+ | 00 | |
| | Quickdraw | –– | –– | –– | 0– | 0– | |
| | Traffic Signs | 0– | 0– | 0– | 0– | 0+ | |
| | VGG Flower | 0– | 0– | 0– | 00 | 00 | |

Table 3: Impact of Paired Testing on Significance Relative to Simple Comparison. In this analysis, we compare the Finetune (FT) method against all other methods across both CLIP and DINO models. The initial character in each pair denotes the significance outcome determined without replacement, while the subsequent character reflects the result derived with paired testing. Here, 0 represents a non-significant difference, + indicates a significantly higher accuracy of the FT method, and – conveys a significantly lower accuracy. The FT columns comparing CLIP and DINO are opposites of each other.

Particularly striking is one instance of inversion, where a method previously deemed significantly more accurate than another was found to be significantly less accurate using a paired test. This reversal was observed in the comparison of Fine-tuning on DINO versus NCC with CLIP features on the Traffic Signs dataset. This implies that a method can significantly outperform another on a specific dataset, yet significantly underperform when evaluated across the entire distribution. The dataset is thus a particular instance of data that favors one method. This example powerfully exemplifies the need for clarity when interpreting CIs.

A natural question that arises from previous considerations is that of how to size tasks when performing sampling without replacement, aiming to reduce the range of obtained CIs. In that matter, we consider the size of the support set to be fixed at $KS$, leaving as the only free variable the number of queries per task and per class $Q$. Indeed, increasing the number of queries will inevitably reduce the total number of tasks we can construct, as shown in the following equation. Assuming a balanced dataset, we can estimate the number of tasks $T$ that can be sampled by exhausting the full dataset:

$$T \approx \left\lfloor \frac{|\mathcal{D}|}{|\mathcal{T}|} \right\rfloor \approx \left\lfloor \frac{CN}{K(Q+S)} \right\rfloor , \tag{6}$$

with $|\mathcal{T}|$ is the number of samples, accounting for both the support and query sets, in each task.

If increasing $Q$ reduces the number of tasks, it also changes $\sqrt{\mathrm{Var}(A_t)}$ which is proportional to the CI. As such, there exists a trade-off between the number of queries and the feasible number of tasks that can be generated to minimize OCIs for any given dataset. Intuitively, measuring $\bar{A}$ with a small $T$ (and consequently a high $Q$) results in extensive CI ranges, a phenomenon depicted in Equation 3. Conversely, measuring with $Q = 1$ may generate many tasks (large $T$) with an extremely high variance because the accuracy per class becomes either 0% or 100%. In the following, we aim to identify the optimal number of queries, denoted $Q^*$,

that effectively minimizes the variance of the average accuracy $\mathrm{Var}(\bar{A})$ and thus the obtained CIs. We first demonstrate mathematically the existence of such minimum by deriving $\mathrm{Var}(\bar{A})$.

## 4 Sizing tasks to narrow OCIs

We show in the Appendix that $\mathrm{Var}(\bar{A})$ can be written as:

$$\mathrm{Var}\left(\bar{A}\right) = \frac{K}{NC}(\alpha Q + \frac{\beta}{Q} + \gamma), \qquad (7)$$

with $\alpha$, $\beta$ and $\gamma$ also defined in the Appendix, and $\beta > 0$.

These parameters are difficult to estimate in particular when dealing with real datasets and methods. If $\alpha \leq 0$, then $\mathrm{Var}\left(\bar{A}\right)$ is decreasing as a function of $Q$ since $Q \in \mathbb{N}$. In the following, we focus only on cases where $\alpha > 0$. This choice is supported by empirical evidence, which we will present later, indicating a U-shaped relationship between the variance of $\bar{A}$ and $Q$ for a certain range of $S$. Assuming this, $\mathrm{Var}\left(\bar{A}\right)$ reaches its minimum at $Q^* = \sqrt{\frac{\beta}{\alpha}}$. Next, we study what this entails as $S$ and $N$ vary.

### 4.1 Effect of $S$ and $N$ on $Q^*$

Given the definition of $\alpha$ and $\beta$ obtained in Equation 10, we find that $Q^*$ is an increasing function of $S$ and a constant function of $N$. In this section, we show that these results are confirmed empirically on real datasets.

We propose to study the aforementioned variance model of $\bar{A}$ with respect to $Q$ in a simplistic 1D representation of samples. In our model, two class' distribution are represented as two Gaussians ($\mathcal{N}_i = \mathcal{N}(\mu_i, \sigma_i)$ with $i \in \{1, 2\}$). We then sample an artificial balanced dataset of fixed size

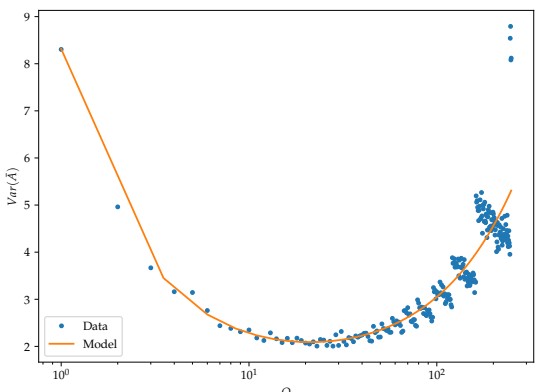

Figure 2: Variance of the average accuracy vs. the number of queries with synthetic data. The two classes are represented as 1D Gaussians $\mathcal{N}(-1, 1)$ and $\mathcal{N}(1, 1)$. The size of the dataset is $N = 1000$ (500 samples per class). Tasks are sampled according to Algorithm 2. The number of shots is set to 5. We fit this with the model described in Equation 10 and observe a strong fit of the model with our experiment.

$N$. Next, we sample tasks within this artificial dataset until exhaustion with the procedure described in Algorithm 2 setting $K = C = 2$. Using the NCC classifier we obtain a set of accuracies on which we can compute the average accuracy $\bar{A}$. This procedure consisting of instantiating the synthetic dataset from Gaussians and measuring $\bar{A}$ is iterated. This yields a set of $\{\bar{A}_j\}_j$ for a given set of parameters $\{S, Q, N, \mathcal{N}_1, \mathcal{N}_2\}$ from which we compute an empirical variance $\mathrm{Var}(\bar{A})$.

In Figure 2, we show the measured $\mathrm{Var}(\bar{A})$ vs $Q$. We use 1000 samples in the datasets, split between the two classes. We set the number of shots to $S = 5$. The model in Equation 10 is fitted very precisely. The discretisation effect seen at high $Q$ is due to the low number of tasks. Next, we study the effect of $S$ and $N$ and compare it to our experiments with synthetic data.

Increasing $S$ shifts the curve's minimum from $Q^* = 1$ towards $Q^* \to +\infty$ as depicted in Figure 3. This aligns with our model's predictions. At $S = 1$, opting for $Q^* = 1$ effectively has two effects: (a) the high variance of $A_t$ due to small support and query sets increases the variance of $\bar{A}$ and (b) low $Q$ allows a significantly larger $T$, thus reducing the overall variance of $\bar{A}$ as shown in Equation 3. Conversely, for $S \geq 20$, the setting boils down to classical transfer learning. Indeed, the narrowest CI is attained with one task with a large support and query set task. Finally, we find a third regime where, for a range of values of $S$, $Q^*$ is nontrivial. It corresponds to what is shown in the $S = 5$ regime in Figure 3.

When increasing $N$, $Q^*$ should not be affected according to Equation 10. However, we observe a hardly perceptible shift of $Q^*$ in Figure 3 when $N$ is increased. We consider this effect to be sufficiently small and therefore negligible. Next, we explore how these results extend to real-world datasets.

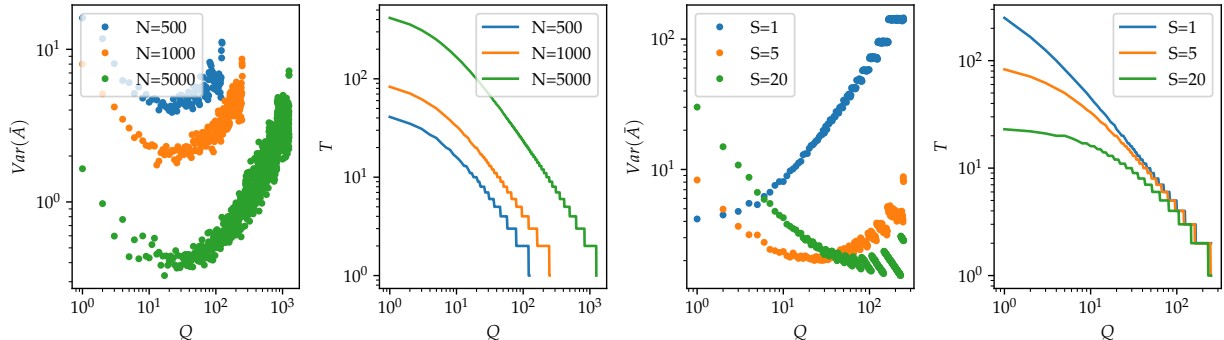

Figure 3: Variance of the average accuracy $\bar{A}$ and number of tasks $T$ across different settings of $S$ and $N$, derived from synthetic datasets featuring two 1D Gaussian classes, $\mathcal{N}(-1, 1)$ and $\mathcal{N}(1, 1)$. The left pair of graphs display results with a fixed number of shots ($S = 5$), while the right pair of graphs show results for a constant sample size in the synthetic dataset ($N = 1000$).

### 4.1.1 Real Dataset Experiments

We now shift our focus to the findings derived from real image datasets. These datasets are often unbalanced and exhibit a variety of class numbers. Our objective is to determine whether the earlier conclusions remain valid in this context.

First, we observe that $Q^*$ does not depend on the size of datasets. However the size of the dataset scales $T$ which in turn scales $CI_{95\%}$. Similar to the results with synthetic data, we observe a discretisation of the confidence interval in the high $Q$ (low $T$) regime.

We also observe a similar phenomenon with different regimes in 1, 5 and 10 shots in Figure 4. Our analysis suggests that for tasks in the 1 shot regime, the best number of queries, $Q$, should be set to 1. For tasks with $S = 5$ and $S = 10$, the best values for $Q$ are approximately $Q = 5$ and $Q = 7$, respectively. Figure 4 also clearly shows that the predominent 15 queries is not optimal to narrow the OCI. We also show similar values for $Q^*$ using DINOv2 instead of CLIP in the Appendix (Figure 5).

## 5 Benchmark Proposal

Building on previously obtained results, we propose a simple benchmark where *Paired Tests* are used and the value of $Q$ is chosen as the minimum found in the previous paragraph. Implicitly, we are assuming that the minimum of $\Delta$'s OCI is also reached at $Q^*$. More precisely, we are assuming the independence of the covariance with respect to $Q$. These assumptions are backed by the improved number of conclusive comparisons in Figure 6 when optimizing $Q$ and using *paired tests*. Indeed, while PT yielded 94 conclusive comparisions, PT with optimized $Q$ yields a little more with 97 conclusive comparisons.

We present our results with the baselines adaptation methods previously studied in Table 6 using DINOv2 as our baseline model. Our experiments consistently show that, for a given model, fine-tuning tends to be less effective than both logistic regression and the nearest class centroid methods. We also observe the choice of model is primordial with a clear advantage of DINOv2 over CLIP and DINO on most datasets. Our benchmark, including the code, seed values, task descriptions, and accuracy results, is available for use.

## 6 Related Work

**Few-Shot Learning** Since the seminal works of Vinyals et al. (2016) and Snell et al. (2017), the field of few-shot learning has known many contributions. Most solutions rely on the use of a pretrained feature extractor, trained on a generic abundant dataset (Wang et al., 2020; Bendou et al.; Antoniou et al., 2018). The feature extractor can then be used as is on the target problem (Wang et al., 2019), or adapted before

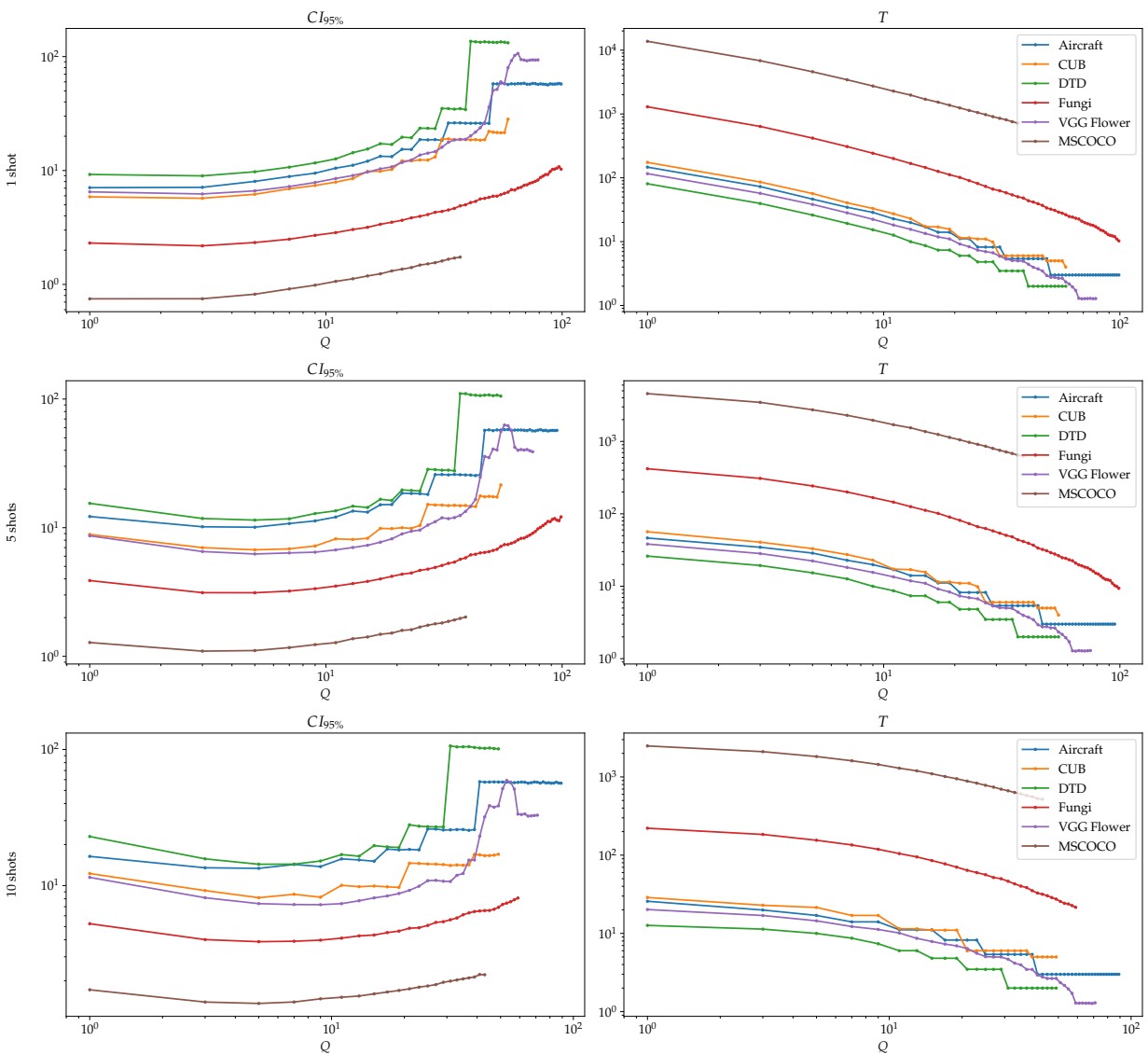

Figure 4: (Left) Confidence Interval ranges (Right) Corresponding number of tasks generated. In all graphs the x-axis is the number of queries $Q$. These results represent averages from multiple trials, with the number of trials tailored according to the Task Count ($T$). Some curves are stopped before $Q$ reaches 100 because of the number of samples per class. We do not show Omniglot and Quickdraw for visibility.

classifying (Zhang et al., 2021). Most proposed methods differ in the way they combine the pretrained feature extractor and their adaptation to the target problem Luo et al. (2023). Over the years, multiple benchmarks have been proposed, including MiniImageNet (Vinyals et al., 2016), Omniglot (Lake et al., 2015) and TieredImegenet (Ren et al., 2018). If initially, these benchmarks focused on the *in-domain* case, where the feature extractor is trained on disjoint classes from the target problem, but all drawn from the same initial dataset, the trend evolved with the introduction of Meta-dataset (Triantafillou et al., 2019) (MD) and later COOP (Zhou et al., 2022a), where the feature extractor is trained on a large generic dataset and applied to various other domains, including fine-grain problems, embodying a *cross-domain* evaluation.

A few papers focus on the sampling of tasks of targeted difficulty for few-shot learning (Arnold et al., 2021; Zhou et al., 2020). In Zhou et al. (2020), the authors claim that model performance can be improved by sampling meta-learning tasks of increasing difficulty. In other works, failed meta-training tasks (deemed hard) are sampled again (Sun et al., 2020) or previously misclassified samples/classes are more likely to be sampled in following tasks Liu et al. (2020) to focus the model on difficult tasks. Estimating task difficulty can itself be a difficult task, and several solutions have been proposed Arnold et al. (2021). The idea of difficulty-based sampling proposed in Arnold et al. (2021) is relevant to this paper since it enables the sampling of groups of tasks with homogeneous difficulty, effectively reducing the confidence interval ranges. In contrast, our research adopts paired tests, a method that obviates the need for such dependencies and provides a more universally applicable approach. Paired tests is not a novel contribution of our work. They were introduced over a century ago to study the evolution of small populations with time (Student, 1908; Hedberg & Ayers, 2015). In these seminal studies, the authors shown that individual differences provide more statistical power and insights than changes in averages over the whole population.

**Confidence Intervals**  Confidence intervals were established by Polish mathematician Jerzy Neyman (Neyman, 1937) in the early 1930's coinciding with Fisher's ideas although supported by a different framework. They were more generally used and then required in medical research around the 1980's. They require assuming a specific model for the distribution of the considered data. In contrast, the bootstrap method, introduced in DiCiccio & Efron (1996), offers a distribution-agnostic approach for estimating ranges. We focused on traditional confidence intervals as they are better understood and more often implemented in the literature of few-shot learning, but similar conclusions could be drawn using Boostrap instead.

**Challenges in Statistical Interpretation and Methodological Biases**  This issue of misleading CIs is not isolated to our domain. Belia et al. (2005) have similarly criticized the common misinterpretations surrounding confidence intervals in broader scientific research. Moreover, the propensity to overlook or underreport negative or null (non-conclusive) results further exacerbates the problem of biased interpretations. Borji (2017) argues for the importance of acknowledging and analyzing negative results in computer vision. Lastly, the impact of dataset biases on the evaluation metrics has been well-documented by Torralba & Efros (2011). In their work, "Unbiased Look at Dataset Biases", they identify and measure several biases such as selection, capture and negative set. It serves as a critical reminder of how dataset-based results can differ from those obtained in real-world distributions.

## 7 Limitations

A first limitation of our study we would like to point out is that for large datasets such as MSCOCO or QuickDraw, the predominant method of CI calculation leads to intervals that are actually larger than our proposed OCI. As such, on such large datasets using CCI may not be an unfair approximation.

Furthermore, our mathematical modeling only explains the origin of the minimum of CI with respect to $Q$ but does not provide a way to find it analytically since we cannot easily estimate $\alpha, \beta$ and $\gamma$ in Equation 10.

As mentioned at the end of paragraph 2.2, saturation at 100% accuracy may negatively impact the computation of confidence intervals and particularly the value of paired tests CI in Equation 5.

Finally, we point out that paired tests introduce complexity as they require a fixed seed and necessitate saving and publishing individual task accuracies when using the benchmark and comparing methods.

## 8 Conclusion

In our study, we demonstrated the stark contrast between Open and Closed measurements of method accuracy in Few-Shot Learning. Notably, OCIs take into account data randomness but are far wider than CCIs. We identified two major approaches that contribute to narrowing the OCIs and subsequently introduced a benchmark which uses these approaches. Our findings underscore the importance of using confidence intervals that account for data randomness in evaluations, a practice we advocate extending beyond classification and vision to encompass all domains employing task-based few-shot learning assessments.

| | Model | CLIP | DINO | DINOv2 | |
|---|---|---|---|---|---|
| Dataset | Method | LR | LR | LR | NCC |
| Aircraft | 1-shot | 9.241 ± 4.274 | 25.931 ± 3.693 | 0.000 ± 1.161 | -0.552 ± 1.091 |
| | 5-shot | 2.286 ± 7.402 | 16.143 ± 6.311 | -2.286 ± 2.043 | -1.286 ± 2.027 |
| | 10-shot | -0.816 ± 5.803 | 10.204 ± 5.083 | -4.286 ± 2.568 | -1.224 ± 2.565 |
| CUB | 1-shot | 10.743 ± 2.067 | 25.486 ± 2.991 | 0.000 ± 0.000 | 0.000 ± 0.000 |
| | 5-shot | 2.545 ± 1.410 | 7.515 ± 2.150 | 0.121 ± 0.247 | -0.121 ± 0.247 |
| | 10-shot | 1.008 ± 1.464 | 4.706 ± 3.114 | 0.000 ± 0.000 | 0.000 ± 0.000 |
| DTD | 1-shot | 7.805 ± 4.722 | 5.122 ± 4.724 | 0.976 ± 1.941 | 0.488 ± 1.191 |
| | 5-shot | 8.750 ± 5.057 | 3.500 ± 4.109 | -0.250 ± 1.979 | 0.000 ± 1.348 |
| | 10-shot | 4.762 ± 2.905 | 1.905 ± 4.392 | -3.492 ± 3.253 | -1.270 ± 1.937 |
| Fungi | 1-shot | 15.357 ± 1.262 | 11.937 ± 1.174 | -0.533 ± 0.507 | 0.235 ± 0.438 |
| | 5-shot | 10.247 ± 1.369 | 3.115 ± 1.225 | -3.098 ± 0.658 | -2.417 ± 0.677 |
| | 10-shot | 7.316 ± 1.286 | 0.584 ± 1.224 | -2.879 ± 0.890 | -1.753 ± 0.799 |
| MSCOCO | 1-shot | 12.360 ± 1.013 | 8.820 ± 0.975 | 0.450 ± 0.377 | 0.180 ± 0.306 |
| | 5-shot | 9.952 ± 0.411 | 6.406 ± 0.389 | -0.196 ± 0.187 | 0.172 ± 0.141 |
| | 10-shot | 8.181 ± 0.371 | 3.883 ± 0.354 | -1.056 ± 0.204 | -0.202 ± 0.158 |
| Omniglot | 1-shot | -1.898 ± 1.227 | -5.923 ± 1.163 | -2.111 ± 0.793 | -0.850 ± 0.755 |
| | 5-shot | 0.760 ± 1.016 | -1.932 ± 0.954 | -1.308 ± 0.564 | -0.973 ± 0.628 |
| | 10-shot | 0.240 ± 1.136 | -2.246 ± 1.027 | -1.788 ± 0.669 | -0.698 ± 0.733 |
| Quickdraw | 1-shot | 4.590 ± 1.079 | 6.530 ± 1.049 | -1.170 ± 0.547 | 1.760 ± 0.526 |
| | 5-shot | 6.016 ± 0.429 | 7.850 ± 0.449 | -1.028 ± 0.210 | -0.160 ± 0.235 |
| | 10-shot | 5.541 ± 0.331 | 6.454 ± 0.354 | -1.109 ± 0.171 | -0.266 ± 0.188 |
| Traffic Signs | 1-shot | -1.770 ± 1.055 | -0.600 ± 1.034 | -0.920 ± 0.519 | 0.520 ± 0.467 |
| | 5-shot | -4.243 ± 0.820 | -4.078 ± 0.673 | -1.537 ± 0.396 | -0.367 ± 0.395 |
| | 10-shot | -5.453 ± 0.948 | -4.919 ± 0.781 | -2.857 ± 0.453 | -0.821 ± 0.443 |
| VGG Flower | 1-shot | 4.576 ± 1.810 | 12.034 ± 2.243 | 0.000 ± 0.477 | 0.000 ± 0.000 |
| | 5-shot | 0.545 ± 0.829 | 2.182 ± 1.521 | 0.000 ± 0.000 | -0.182 ± 0.378 |
| | 10-shot | 0.000 ± 0.000 | 1.039 ± 0.968 | 0.260 ± 0.579 | 0.260 ± 0.579 |

Table 4: Comparative differences in paired tests. This table contrasts the performance of DINOv2 with Fine-tuning (FT) against DINOv2 combined with Nearest Class Centroid (NCC) or Logistic Regression (LR), as well as the performance combinations of CLIP with DINO using LR.

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

# A  Mathematical derivation of $\mathrm{Var}(\bar{A})$

Suppose tasks are drawn IID *without* replacement, we write the variance of $\bar{A}$ as:

$$\mathrm{Var}\left(\bar{A}\right) = \frac{1}{T}\mathrm{Var}_t\left(A_t\right), \tag{8}$$

with $A_t$ the accuracy for an arbitrary task $t$. By definition of the variance,

$$\mathrm{Var}\left(A_t\right) = \mathbb{E}[(A_t)^2] - (\mathbb{E}[A_t])^2. \tag{9}$$

For a given support set, the expected accuracy for some class $c$ is denoted $\mu_{t,c}$.

$$\mu_{t,c} \triangleq \mathbb{E}_{\mathcal{Q}_t}(\mathbb{1}[f_{\mathcal{S}_t}(x) = c] \mid \mathcal{S}_t).$$

Then the expectation of $A_t$ becomes

$$\mathbb{E}[A_t] = \mathbb{E}_{\mathcal{S}_t}[\mu_{t,c}],$$

Along the same lines, we can derive $\mathbb{E}[(A_t)^2]$,

$$\mathbb{E}[(A_t)^2] = \mathbb{E}_{\mathcal{S}_t}\mathbb{E}_{\mathcal{Q}_t}[(A_t)^2|\mathcal{S}_t].$$

For a fixed $S_t$, $\mathbb{1}[f_{\mathcal{S}_t}(x) = c]$ and $\mathbb{1}[f_{\mathcal{S}_t}(x') = c']$ are not independent and their distribution will depend on the classes $c$ and $c'$. Using Equation 1, we obtain .

$$\mathbb{E}[(A_t)^2] = \frac{1}{(KQ)^2}\sum_c\sum_{c'}\sum_x\sum_{x'}\mathbb{E}_{\mathcal{S}_t}\mathbb{E}_{\mathcal{Q}_t}[\mathbb{1}[f_{\mathcal{S}_t}(x) = c]\mathbb{1}[f_{\mathcal{S}_t}(x') = c']].$$

We now separate cases where $x = x'$ from cases where $c = c'$ and finally cases where both are different.

$$\mathbb{E}[(A_t)^2] = \frac{1}{KQ}\mathbb{E}_{\mathcal{S}_t}[\mu_{t,c}] + \frac{Q-1}{KQ}\mathbb{E}_{\mathcal{S}_t}[(\mu_{t,c})^2] + \frac{1}{K^2}\sum_c\sum_{c'\neq c}\mathbb{E}_{\mathcal{S}_t}[\mu_{t,c}\mu_{t,c'}].$$

Using Equation 9, we find:

$$\mathrm{Var}\left(A_t\right) = \frac{1}{KQ}\mathbb{E}_{\mathcal{S}_t}[\mu_{t,c}] + \frac{Q-1}{KQ}\mathbb{E}_{\mathcal{S}_t}[(\mu_{t,c})^2] + \frac{1}{K^2}\sum_c\sum_{c'\neq c}\mathbb{E}_{\mathcal{S}_t}[\mu_{t,c}\mu_{t,c'}] - (\mathbb{E}_{\mathcal{S}_t}[\mu_{t,c}])^2.$$

Let us define some parameters,

$$m_1 \triangleq \mathbb{E}[A_t] = \frac{1}{K}\sum_c\mathbb{E}_{\mathcal{S}_t}[\mu_{t,c}],$$

$$m_2 \triangleq \frac{1}{K}\sum_c\mathbb{E}_{\mathcal{S}_t}[(\mu_{t,c})^2],$$

and

$$m_3 \triangleq \frac{1}{K^2}\sum_c\sum_{c'\neq c}\mathbb{E}_{\mathcal{S}_t}[\mu_{t,c}\mu_{t,c'}].$$

We get that

$$\mathbb{E}[(A_t)^2] = \frac{1}{KQ}m_1 + \frac{Q-1}{KQ}m_2 + m_3.$$

This gives

$$\mathrm{Var}\left(A_t\right) = \frac{1}{KQ}m_1 + \frac{Q-1}{KQ}m_2 + m_3 - (m_1)^2.$$

Then, using Equation 6 (removing the rounding) and 8, we approximate:

$$\mathrm{Var}\left(\bar{A}\right) = \frac{K}{NC}(\alpha Q + \frac{\beta}{Q} + \gamma), \tag{10}$$

with $\alpha = \frac{m_2}{K} + m_3 - (m_1)^2, \beta = \frac{S}{K}(m_1 - m_2)$ and $\gamma = \frac{m_1}{K} - \frac{m_2}{K} + \frac{S}{K}m_2 + S(m_3 - (m_1)^2)$.

First, let us notice that $\beta > 0$ since for $\mu \in [0, 1], \mu^2 < \mu$ .

## B  Algorithms

Algorithm 1 samples tasks with replacement and computes CCIs with Equation 3 while Algorithm 2 samples tasks without replacement and uses the student's distribution to compute OCIs.

---

**Algorithm 2** Full dataset no-replacement evaluation algorithm

---

 1: **procedure** EVALUATEUNTILDEPLETED$(K, S, Q, C, \{\mathcal{X}_c\}_{c \in \mathcal{C}})$  ▷ $K$ ways, $S$ shots, $Q$ queries,
    set of classes $C$, set of data samples $\{\mathcal{X}_c\}_{c \in \mathcal{C}}$
 2:    Initialize $\mathcal{T} = \{\}$                                    ▷ List to store all tasks
 3:    **while** There are at least $K$ classes in $\mathcal{C}$ with at least $S + Q$ examples each **do**
 4:        $\mathcal{K} \leftarrow$ Randomly select $K$ classes from $\mathcal{C}$ with at least $S + Q$ examples
 5:        Initialize $\mathcal{S} = \{\}$ and $\mathcal{Q} = \{\}$
 6:        **for** each $c$ in $\mathcal{K}$ **do**
 7:            $\mathcal{S}_c \leftarrow$ Randomly select $S$ examples from $\mathcal{X}_c$
 8:            $\mathcal{Q}_c \leftarrow$ Randomly select $Q$ examples from $\mathcal{X}_c$ excluding $\mathcal{S}_c$
 9:            Remove $\mathcal{S}_c$ and $\mathcal{Q}_c$ from $\mathcal{X}_c$
10:            Add $\mathcal{S}_c$ to $\mathcal{S}$
11:            Add $\mathcal{Q}_c$ to $\mathcal{Q}$
12:        **end for**
13:        Add $(\mathcal{S}, \mathcal{Q})$ to $\mathcal{T}$
14:    **end while**
15:    Initialize $\mathcal{A} = \{\}$                                  ▷ List to store all accuracies
16:    **for** $t \in \mathcal{T}$ **do**
17:        Add $A_t$ to $\mathcal{A}$                                   ▷ Measure the accuracy of task t
18:    **end for**
19:    $\bar{A} = \mathrm{MEAN}(\mathcal{A})$
20:    $\delta_{95\%} = t(|\mathcal{A}| - 1, 95\%)\sqrt{\frac{Var(\mathcal{A})}{|\mathcal{A}|}}$     ▷ t is the critical value for the Student't distribution
21:    **return** $\bar{A} \pm \delta_{95\%}$
22: **end procedure**

---

## C   Additional results on the benchmark

These results show the performance differences when taking DINO and CLIP with finetune as baselines. Again finetuning mostly underperforms other adaptation methods.

| Dataset | Method Sampling | LR | NCC |
|---|---|---|---|
| Aircraft | 1-shot | -0.690 ± 1.790 | 0.690 ± 1.519 |
| | 5-shot | -5.286 ± 2.802 | -2.857 ± 2.731 |
| | 10-shot | -5.510 ± 3.391 | -0.408 ± 3.290 |
| CUB | 1-shot | -1.486 ± 1.467 | 0.914 ± 1.388 |
| | 5-shot | -2.182 ± 1.740 | -0.970 ± 1.738 |
| | 10-shot | -4.874 ± 2.737 | -3.529 ± 2.822 |
| DTD | 1-shot | -1.220 ± 2.111 | -0.488 ± 1.540 |
| | 5-shot | -1.750 ± 2.457 | -0.250 ± 2.518 |
| | 10-shot | -2.540 ± 1.320 | -1.587 ± 1.937 |
| Fungi | 1-shot | -0.549 ± 0.561 | 1.427 ± 0.514 |
| | 5-shot | -3.166 ± 0.728 | -1.770 ± 0.627 |
| | 10-shot | -3.658 ± 0.920 | -3.009 ± 0.789 |
| MSCOCO | 1-shot | 0.840 ± 0.389 | 0.000 ± 0.380 |
| | 5-shot | -0.846 ± 0.199 | -0.210 ± 0.176 |
| | 10-shot | -2.488 ± 0.224 | -0.695 ± 0.193 |
| Omniglot | 1-shot | 2.976 ± 0.675 | 2.749 ± 0.724 |
| | 5-shot | 1.430 ± 0.484 | -0.183 ± 0.556 |
| | 10-shot | 0.065 ± 0.560 | -0.458 ± 0.656 |
| Quickdraw | 1-shot | 1.800 ± 0.646 | 3.590 ± 0.691 |
| | 5-shot | -0.546 ± 0.267 | -0.680 ± 0.303 |
| | 10-shot | -1.841 ± 0.223 | -1.531 ± 0.260 |
| Traffic Signs | 1-shot | 0.880 ± 0.429 | 1.310 ± 0.486 |
| | 5-shot | -0.900 ± 0.354 | 0.911 ± 0.372 |
| | 10-shot | -2.189 ± 0.400 | 0.725 ± 0.412 |
| VGG Flower | 1-shot | -2.542 ± 1.755 | 1.695 ± 1.475 |
| | 5-shot | -0.545 ± 0.623 | -0.909 ± 1.085 |
| | 10-shot | -0.519 ± 0.776 | -0.519 ± 0.776 |

Table 5: Paired test difference between DINO with FT and LR and NCC on DINO. FT, NCC and LR respectively stand for Fine-tuning, Nearest Class Centroid, Logistic Regression.

| Dataset | Method Sampling | LR | NCC |
|---|---|---|---|
| Aircraft | 1-shot | $0.828 \pm 1.850$ | $4.000 \pm 2.033$ |
| | 5-shot | $0.000 \pm 1.934$ | $0.571 \pm 2.301$ |
| | 10-shot | $-3.469 \pm 2.436$ | $-5.510 \pm 2.191$ |
| CUB | 1-shot | $-0.229 \pm 1.500$ | $1.714 \pm 1.313$ |
| | 5-shot | $-0.364 \pm 0.962$ | $-0.848 \pm 1.266$ |
| | 10-shot | $-0.504 \pm 0.934$ | $-0.504 \pm 1.297$ |
| DTD | 1-shot | $-2.195 \pm 2.184$ | $2.195 \pm 2.584$ |
| | 5-shot | $-2.500 \pm 3.002$ | $-2.000 \pm 2.806$ |
| | 10-shot | $-5.079 \pm 2.859$ | $-4.444 \pm 1.937$ |
| Fungi | 1-shot | $-2.086 \pm 0.747$ | $0.439 \pm 0.703$ |
| | 5-shot | $-3.217 \pm 0.792$ | $-2.826 \pm 0.868$ |
| | 10-shot | $-4.394 \pm 0.929$ | $-4.069 \pm 0.986$ |
| MSCOCO | 1-shot | $-0.840 \pm 0.536$ | $1.670 \pm 0.528$ |
| | 5-shot | $-2.090 \pm 0.241$ | $-0.970 \pm 0.232$ |
| | 10-shot | $-3.099 \pm 0.239$ | $-1.503 \pm 0.233$ |
| Omniglot | 1-shot | $4.723 \pm 0.725$ | $4.723 \pm 0.799$ |
| | 5-shot | $1.688 \pm 0.606$ | $0.532 \pm 0.670$ |
| | 10-shot | $0.371 \pm 0.702$ | $-0.087 \pm 0.831$ |
| Quickdraw | 1-shot | $0.760 \pm 0.731$ | $4.940 \pm 0.759$ |
| | 5-shot | $0.226 \pm 0.299$ | $-0.284 \pm 0.327$ |
| | 10-shot | $-0.376 \pm 0.236$ | $-0.754 \pm 0.268$ |
| Traffic Signs | 1-shot | $-0.500 \pm 0.590$ | $0.590 \pm 0.606$ |
| | 5-shot | $-0.481 \pm 0.373$ | $-0.745 \pm 0.449$ |
| | 10-shot | $-1.642 \pm 0.418$ | $-0.458 \pm 0.478$ |
| VGG Flower | 1-shot | $-1.356 \pm 1.408$ | $0.169 \pm 1.305$ |
| | 5-shot | $-0.182 \pm 0.665$ | $0.000 \pm 0.774$ |
| | 10-shot | $-0.519 \pm 0.776$ | $-0.519 \pm 0.776$ |

Table 6: Paired test difference between CLIP with FT and LR and NCC on CLIP. FT, NCC and LR respectively stand for Fine-tuning, Nearest Class Centroid, Logistic Regression.

# D    Is $Q^*$ Dependent on the Model Used?

We show in Figure 5 that the same values of $Q^*$ are found for DINO v2 instead of CLIP.

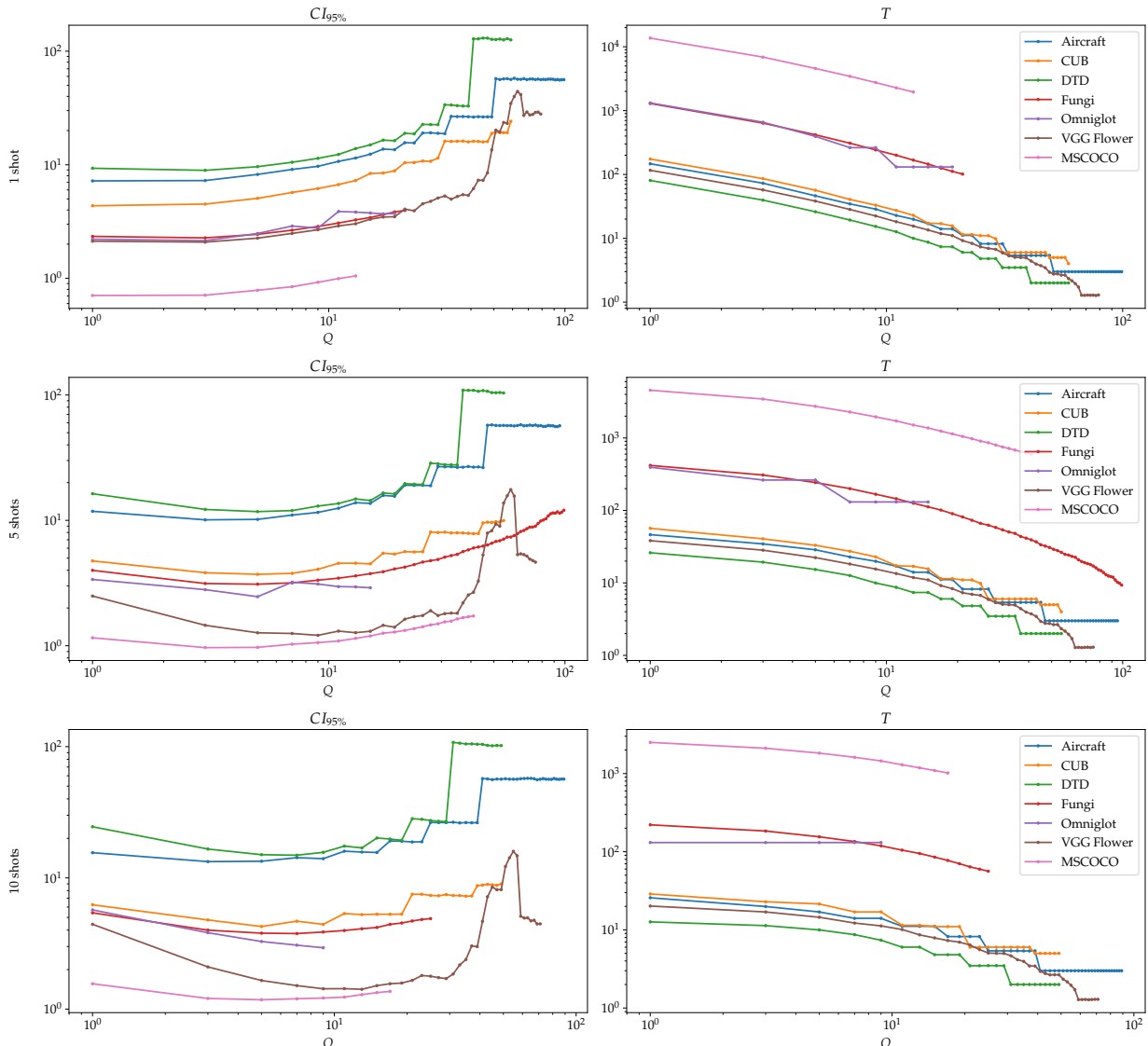

Figure 5: This figure shows that the value of $Q^*$ is not dependent on the model used. This experiment is conducted using DINO v2 instead of CLIP. (Left) Confidence Interval ranges (Right) Corresponding number of tasks generated. In all graphs the x-axis is the number of queries $Q$. These results represent averages from multiple trials on DINO v2, with the number of trials tailored according to the Task Count ($T$). Some curves are stopped before $Q$ reaches 100 because of the number of samples per class. We do not show Omniglot and Quickdraw for visibility.

# E   Statistics on Conclusiveness

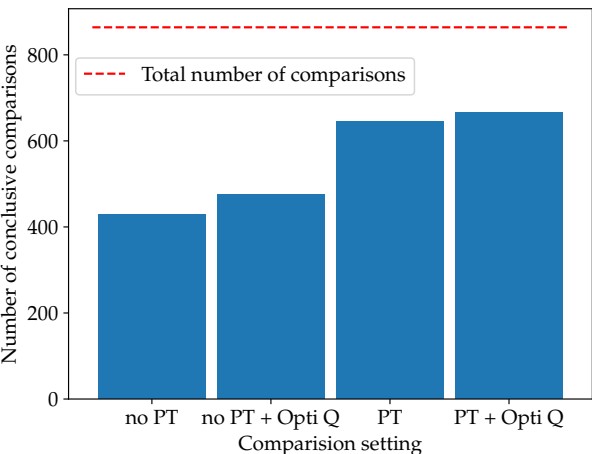

Figure 6: Effect of using paired tests comparisons between all possible pairs formed from combinations of models and methods summed across datasets (with *Omniglot* removed for technical reasons) for 1, 5 and 10 shots. The number of conclusive comparisons is counted on the Y-axis. When $Q$ is not optimized, we use $Q = 15$.

This histogram illustrates that *paired test* and the optimization of the size of tasks yields the maximum number of conclusive comparisons. Optimizing $Q$ slightly improves the number of conclusive comparisons compared to simple paired tests.

## Acknowledgment

This work was performed using HPC resources from GENCI–IDRIS (Grant 202123656B).

The PhD study of Luke Smith is supported by GSK-plc and the Australian Government Research Training Program (RTP) scholarship.

Funded by the Deutsche Forschungsgemeinschaft (DFG, German Research Foundation) under Germany's Excellence Strategy EXC 2044 –390685587, Mathematics Münster: Dynamics–Geometry–Structure.

F. Vermet conducted this work within the France 2030 framework programme, the Centre Henri Lebesgue ANR-11-LABX-0020-01

Avec le soutien de

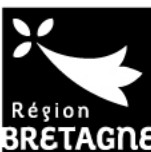

Raphael Lafargue was supported by Brittany Region, France.

