# OpenReview forum: "Oops, I Sampled it Again: Reinterpreting Confidence Intervals in Few-Shot Learning"
_TMLR — Accepted by TMLR_

### Review · Reviewer_pXGx · 2024-06-09

**Summary Of Contributions:**

This paper investigates the problems of computing confidence intervals for few-shot learning models. Computing confidence intervals is very important as the performance of few-shot models can vary greatly on different sampled tasks.

The predominant approach allows the same examples to appear in multiple tasks when sampling the tasks (with replacement), named Closed CIs (CCIs) in this paper. Although more tasks can be sampled in this way, CCIs may overfit on specific datasets and not reflect models' performance on real distribution. Open CIs (OCIs), which sample tasks without replacement, are more representative of quantifying models' ability. This paper shows that OCIs are much larger than CCIs, especially in datasets with fewer examples, and their difference is more substantial. Compared to CCIs, using OCIs can lead to completely inverse conclusions when comparing few-shot methods.

Motivated by the correlation of models' accuracies on the same task, this paper proposes the Paired Test (PT), which lets two models evaluate the same task and compute their performance gaps. Pair Tests can reduce the confidence intervals, and the conclusions drawn from PT do not contradict those drawn from OCI. In addition, this paper investigates the influence of query size Q on CI and proposes a way to find the optimal Q. Finally, based on all those findings, this paper introduces a benchmark for future evaluation of few-shot learning methods.

**Audience:**

Yes

**Broader Impact Concerns:**

None.

**Claims And Evidence:**

Yes

**Requested Changes:**

1. Have a limitation section to discuss the scope of usage of the proposed method.
2. The writing could be gentler, making it easier for readers to digest the information. For example, Table 1 contains too much information, and until the very end of the introduction, this table is discussed. Consider moving it to page 2 or adding a paragraph to describe the results earlier.

**Strengths And Weaknesses:**

Strengths:
1. **Important findings and well-motivated problems.** This paper finds that the existing evaluation methods for few-shot learning are problematic. Different evaluation methods can lead to different conclusions. Having reliable evaluation metrics is the foundation for developing algorithms. The problems this paper studied are very important for more rigorous research in few-shot learning.

2. **Simple yet intuitive methods.** The paired test is intuitive but effective in narrowing the interval while maintaining the number of tasks.

3. **Comprehensive experiments.** This paper evaluates various datasets, and the comprehensive experiments and ablations support the claims.

4. The paper is, in general, easy to follow. The writing tells the story well.

Weakness
1. **Application on large-scale datasets.** It looks like the large interval problems only happen in small datasets. For large datasets (shown in Table 2), the OCIs are even smaller than CCIs. In this case, the proposed method may not be necessary for large datasets.

2. **The choice of optimal Q.** If I understand correctly, the empirical choice of Q depends on a specific model, which could be biased. This Q* you find may not be the global optimal size. Given the results in Figure 5, the improvement of using Optimized Q is marginal, making me skeptical about whether this Optimized Q is actually helpful.

To be honest, I am not very familiar with this paper's background literature, so my comments may not be precise.

---

> ### Author Response · Authors · 2024-07-12
>
> We would like to thank the reviewer for their valuable feedback. We are pleased to know that they found our work to be important, comprehensive, and easy to follow. We now address the identified weaknesses and requested changes.
>
> # Weaknesses
>
> 1) Indeed, this study found that the discrepancy CCI << OCI occurs on small datasets only, which are relatively common in FSL benchmarks. It is not surprising that large datasets do not exhibit the same limitations since more than 600 tasks can be sampled from them without replacement. In MetaDataset, only MSCOCO and Quickdraw are “large enough”, whereas all other datasets are not.
>
> 2) Indeed, the optimal value of Q* might be sensitive to the model used. We ran experiments that we added to the appendix showing very similar results for the value of Q* using DINO v2 instead of CLIP (in the Appendix: Figure 5). We agree that the improvements due to the optimization of Q, as shown in Figure 5, are marginal. Yet we point out that the main purpose of the paper was to properly and soundly address these questions, and that the fact Q* do not bring significant improvements and remain in our view a valid and interesting finding of our work.
>
> # Requested Changes:
>
> 1) We added a  limitation section that includes both points above. We also discuss the limitation of our theoretical model, and the saturation of accuracy at 100% which can limit the covariance term in Equation 5.
>
> 2) We made the text easier to digest by updating the transitions, cropping (removing LR from the table), and moving Table 1 to page 2.

---

> > ### Comment · Reviewer_pXGx · 2024-07-15
> > **Response to authors' comments**
> >
> > Thanks for your comments which addressed my concerns.

---

### Review · Reviewer_oQ3o · 2024-06-15

**Summary Of Contributions:**

The paper considers the problem of computing confidence intervals (CIs) in few-shot learning (FSL). The key observation that drives the paper is that the common way of computing CIs, which is based on sampling with replacement, can lead to misleading results as it does not take sufficiently into account the randomness of the data. To quantify that discrepancy, they compare the results obtained by computing CIs without replacement. They find that many results that reported improvements in the literature are in fact inconclusive when sampling tasks without replacement. Finally, they propose ways of reducing the range of the CI by sampling tasks with a specific size. Notably, they rely on the known method of so-called paired evaluations in order to obtain more conclusive results.

**Audience:**

Yes

**Broader Impact Concerns:**

I do not have any concerns on the ethical implications of this work.

**Claims And Evidence:**

Yes

**Requested Changes:**

Some minor comments which are not crucial for my recommendation:
- The citation style is used incorrectly throughout; for example, "any assertions about this method contingent upon a specific
set of hyperparameters Kumar et al. (2022)" should instead read any assertions about this method contingent upon a specific
set of hyperparameters (Kumar et al. 2022). That is, use paranethesis when the citation is not syntantically part of the sentence.
- In the beginning of page 9, there is typo in the sentence " We consider this effect to be to small enough"
- First equation in page 16 is missing a punctation mark.

**Strengths And Weaknesses:**

Overall, I believe that the paper tackles a crucial problem at the heart of evaluating FSL models, which is typically ignored in this line of work. They provide extensive experimental evaluations supporting the fact that there is a significant discrepancy between closed and open CIs. Indeed, they found that, across various datasets, CCIs can be almost 4 times larger than OCIs. The authors also propose ways to reduce the range of the CI by using i) paired tests, and ii) suitably sizing the tasks through a principled framework.

The results have a bearing in existing, reported results in prior literature, and highlight an important consideration for the literature going forward. Thus, although I am not an expert in this area, I believe that this paper could have significant impact. I also found the evaluation quite thorough and complete in terms of using the existing available datasets. The writing overall is very clear and well-organized, apart from some minor comments stated below.

One weakness of the paper is that at times the main body is not self-contained without the appendix, and important aspects are missing; for example, it would be nice if Algorithm 1 was in the main body, and Section 4 can also be improved in that regard, although I understand that the constraint on the page limit could make this difficult.

---

> ### Author Response · Authors · 2024-07-12
>
> We would like to thank the reviewer for their valuable feedback. We are pleased to know that they found this paper to be impactful, well supported, and relevant for the readership of TMLR.
>
> We updated the paper according to all your requested changes. Regarding your comments on the self-containment of the paper, we asked the Action Editors if we could exceed the 12 page limit (fast review/decision process). Based on their response, we decided to include Alg. 1 in the main body but keep Alg. 2 and the mathematical derivations in the Appendix.

---

> > ### Comment · Reviewer_oQ3o · 2024-07-15
> >
> > I thank the authors for the response. My comments have been adequately addressed.

---

### Review · Reviewer_ZiuS · 2024-07-04

**Summary Of Contributions:**

This paper aims to address the issue of computing confidence intervals (CIs) in few-shot learning (FSL) by sampling tasks with replacement. This predominant method results in misleading CIs that account for the randomness of the sampler but not the data itself. In particular, this paper first highlights the issue of the current CI computation and shows that computing CIs with replacement would underestimate the true variability. Then, this paper develops a paired test, that is evaluated on the same set of tasks, to address the aforementioned issue. Accordingly, this paper also develops a new benchmark for evaluating the performance of few-shot learning.

**Audience:**

Yes

**Broader Impact Concerns:**

No concern

**Claims And Evidence:**

Yes

**Requested Changes:**

See the weakness part.

**Strengths And Weaknesses:**

Strengths:

* The paper identifies a fundamental flaw in the standard method of computing CIs in FSL, which is crucial for the validity of comparative studies in this field.
* The authors conduct a thorough comparative analysis between CIs computed with and without replacement, demonstrating the extent of the problem.
* The proposal of paired tests and optimized task sampling methods offers practical solutions to improve the accuracy and reliability of CI computations in FSL.
* The paper provides empirical evidence to support the proposed methods, using standard FSL benchmarks to illustrate the improvements.  The paper also develops a new benchmark for FSL.

Weaknesses:
* The scope of this paper is a bit limited, the empirical validation is primarily focused on vision datasets in FSL. It remains to be seen how well these methods generalize to other types of datasets and tasks.
* The effectiveness of the proposed methods may depend on specific conditions, such as the size and composition of the datasets, which might limit their applicability in diverse scenarios.
* Theoretical analysis may be lacking. Almost all results are interpreted via empirical study, the underlying theoretical explanation is not clear.
* The complexity of implementing the paired test should be discussed.

---

> ### Author Response · Authors · 2024-07-12
>
> We would like to thank the reviewer for their valuable feedback.  We are pleased that they found our work useful and important. In the following, we answer your concerns about the paper.
>
> ## Weaknesses
>
> > The scope of this paper is a bit limited, the empirical validation is primarily focused on vision datasets in FSL. It remains to be seen how well these methods generalize to other types of datasets and tasks.
>
> We believe our work to be primarily theoretical, with empirical demonstrations on vision datasets. Indeed, it applies exclusively to Few-Shot Learning (FSL) for the reasons mentioned in the introduction: (a) a large number of tasks is required due to the small training/support set in FSL, as a single task would be highly biased and thus unfair for assessing an FSL method, and (b) datasets are size-limited. We better highlighted this point in the introduction. We chose to focus on vision as we believed the most common benchmarks not only suffer from the fundamental flaw of using CCI, but also draw conclusions that would become inconclusive if not.
> We agree that it is left as future work to see how these findings could be extended to other tasks and modalities, and added comments about it into a new limitation section.
>
>
> > The effectiveness of the proposed methods may depend on specific conditions, such as the size and composition of the datasets, which might limit their applicability in diverse scenarios.
>
> Indeed, the effects of the dataset size and composition on the CI are important. We also added a discussion on that point in the limitation section.
>
> > Theoretical analysis may be lacking. Almost all results are interpreted via empirical study, the underlying theoretical explanation is not clear
>
> In the Appendix, we offer a detailed derivation demonstrating how variations in the query set size impact the CI. Additionally, we provide a theoretical explanation of how paired tests can help reduce the CI. We would be grateful for any feedback on aspects that may be unclear or need further clarification, and we are happy to provide additional explanations as needed.
> > The complexity of implementing the paired test should be discussed.
>
> Paired tests introduce complexity as they require a fixed seed and necessitate saving and publishing individual task accuracies when using the benchmark and comparing methods. It is indeed a point that was worth mentioning in the paper. The reviewer will find a clarification on that aspect in our updated version of the manuscript.

---

### Decision · Action_Editor_Rfsy · 2024-08-20

**Recommendation:** Accept as is

**Comment:**

This paper conducts an investigation into the evaluation methodology for few-shot classification tasks. Specifically, they identify a flaw in the procedure with which confidence intervals are computed: they take into consideration only the randomness of the sampler but not the data. This is due to how 'tasks' are sampled to evaluate a few-shot learning method, where examples are sampled with replacement (leading to the same samples appearing in multiple 'tasks'), thus producing 'closed CIs'. Instead, one could have computed 'Open CIs' by sampling without replacement, at the cost of having many fewer tasks available (and larger confidence intervals), which is problematic in few-shot learning datasets that are typically small. The authors propose strategies to address this including a Paired Test where different methods are evaluated on the same set of sampled tasks. This allows to perform meaningful comparisons and reach conclusive conclusions easier than in Open CIs due to reducing the confidence intervals.

As all reviewers pointed out, the paper studies an important problem and the claims and findings in this paper are insightful and can be very impactful for how few-shot learning success is measured moving forward, e.g. "Overall, I believe that the paper tackles a crucial problem at the heart of evaluating FSL models, which is typically ignored in this line of work." (Reviewer oQ3o) and "The problems this paper studied are very important for more rigorous research in few-shot learning." (Reviewer pXGx).

The reviewer also appreciated the quality of the writing ("The writing overall is very clear and well-organized", Reviewer oQ3o) and, as discussed above in the "claims and evidence" section, all reviewers found the claims well-supported, and the empirical investigation to be thorough. Overall, I view this paper as a nice contribution for the field of few-shot learning. The findings reveal interesting and previously-unknown conclusions and can inform how few-shot evaluation is carried out moving forward.

A limitation seems to be that the empirical investigation is focused primarily on small datasets (which is justifiable as these are common in few-shot learning benchmarks), but it is not clear if the issues identified here appear in larger datasets as well, and how the proposed methods fare in that case.

**Audience:**

Yes, this paper is on the important topic of (the evaluation of) few-shot learning, which is of interest to the TMLR audience.

**Claims And Evidence:**

This paper conducts an investigation into the evaluation methodology for few-shot classification tasks. Specifically, they identify a flaw in the procedure with which confidence intervals are computed: they take into consideration only the randomness of the sampler but not the data. This is due to how 'tasks' are sampled to evaluate a few-shot learning method, where examples are sampled with replacement (leading to the same samples appearing in multiple 'tasks'), thus producing 'closed CIs'. Instead, one could have computed 'Open CIs' by sampling without replacement, at the cost of having many fewer tasks available (and larger confidence intervals), which is problematic in few-shot learning datasets that are typically small. The authors propose strategies to address this including a Paired Test where different methods are evaluated on the same set of sampled tasks. This allows to perform meaningful comparisons and reach conclusive conclusions easier than in Open CIs due to reducing the confidence intervals.

The argument of this work is mostly theoretical, and the reviewers pointed out that it was clearly written. The authors based their claims on empirical investigation that, while limited to classification tasks, the reviewers found thorough. Specifically, the reviewers pointed out that the authors "conduct a thorough comparative analysis between CIs computed with and without replacement, demonstrating the extent of the problem" (Reviewer ZiuS) and "The paper provides empirical evidence to support the proposed methods, using standard FSL benchmarks to illustrate the improvements" (Reviewer ZiuS); "They provide extensive experimental evaluations supporting the fact that there is a significant discrepancy between closed and open CIs." (Reviewer oQ3o) and "This paper evaluates various datasets, and the comprehensive experiments and ablations support the claims." (Reviewer pXGx).